# Bioengineered human myobundles mimic clinical responses of skeletal muscle to drugs

Lauran Madden[1], Mark Juhas[1], William E Kraus[2], George A Truskey[1], Nenad Bursac[1]*

[1]Department of Biomedical Engineering, Duke University, Durham, United States; [2]Department of Medicine, Duke University School of Medicine, Durham, United States

**Abstract** Existing in vitro models of human skeletal muscle cannot recapitulate the organization and function of native muscle, limiting their use in physiological and pharmacological studies. Here, we demonstrate engineering of electrically and chemically responsive, contractile human muscle tissues ('myobundles') using primary myogenic cells. These biomimetic constructs exhibit aligned architecture, multinucleated and striated myofibers, and a Pax7+ cell pool. They contract spontaneously and respond to electrical stimuli with twitch and tetanic contractions. Positive correlation between contractile force and GCaMP6-reported calcium responses enables non-invasive tracking of myobundle function and drug response. During culture, myobundles maintain functional acetylcholine receptors and structurally and functionally mature, evidenced by increased myofiber diameter and improved calcium handling and contractile strength. In response to diversely acting drugs, myobundles undergo dose-dependent hypertrophy or toxic myopathy similar to clinical outcomes. Human myobundles provide an enabling platform for predictive drug and toxicology screening and development of novel therapeutics for muscle-related disorders.

*For correspondence: nbursac@duke.edu

**Competing interests:** The authors declare that no competing interests exist.

## Introduction

Development of human in vitro systems for basic biological studies and drug discovery is motivated by the need to improve outcomes in human patients and alleviate ethical considerations demanding a reduction in the use of animals (*Dambach and Uppal, 2012*; *Bhatia and Ingber, 2014*). While significant progress has been made towards predictive in vitro models for liver, lung, and cardiac tissues (*Bhatia and Ingber, 2014*), a functional model of human skeletal muscle has not been described. This is of particular concern as there are a wide range of metabolic, neuromuscular, and dystrophic disorders involving skeletal muscle that are under investigation and still lacking therapies. Skeletal muscle is also central to diseases with high societal impact and those that do not have adequate animal models, including diabetes, obesity, and different dystrophies. Furthermore, through secretion of contraction-dependent myokines, skeletal muscle has been strongly implicated in organ–organ interactions including processes as diverse as cognition, inflammation, cancer, and aging (*Pedersen and Febbraio, 2012*). The need for an accurate preclinical model of human skeletal muscle was exemplified by the market withdrawal of cerivastatin that was well tolerated in mice but caused fatal rhabdomyolysis in humans (*von Keutz and Schluter, 1998*; *Thompson et al., 2006*).

Expansion of primary human myoblasts and formation of myotubes in two-dimensional (2D) systems is well known, however, these cultures are difficult to maintain over long times, lack the architecture of native muscle, and require complex media components to initiate spontaneous contractions (*Blau and Webster, 1981*; *Eberli et al., 2009*; *Guo et al., 2014*). The contractile force of single, in vitro cultured human myofibers can be measured (*Smith et al., 2014*), though such a system is limited by its inability

**eLife digest** Scientists have developed realistic models of the human liver, lung, and heart that allow them to observe living tissue in the laboratory. These models have helped us to better understand how these organs work and what goes wrong in diseases that affect these organs. The models can also be used to test how new drugs may affect a particular organ without the risk of exposing patients to the drug.

Efforts to develop a realistic laboratory model of human muscle tissues that can contract like real muscles have not been as successful to date. This shortcoming has potentially hindered the development of drugs to treat numerous disorders that affect muscles and movement in humans—such as muscular dystrophies, which are diseases in which people progressively lose muscle strength.

Some important drugs, like cholesterol-lowering statins, have detrimental effects on muscle tissue; one statin was so harmful to muscles that it had to be withdrawn from the market. As such, it would be useful to have experimental models that would allow scientists to test whether potential drugs damage or treat muscle tissue.

Madden et al. have now bioengineered a three-dimensional laboratory model of living muscle tissue made of cells taken from biopsies of several different human patients. These tissues were grown into bundles of muscle fibers on special polymer frames in the laboratory. The bioengineered muscle bundles respond to electrical and chemical signals and contract just like normal muscle. They also exhibit the same structure and signaling as healthy muscle tissue in humans.

Madden et al. exposed the muscle tissue bundles to three drugs known to affect muscles to determine if the model could be used to test whether drugs have harmful effects. This revealed that the bundles had weaker contractions in response to statins and the malaria drug chloroquine, just like normal muscles do—and that this effect worsened if more of each drug was used. Madden et al. also found that a drug that strengthens muscle contractions at low doses and damages muscle at high doses in humans has similar effects in the model.

As well as this model being used to screen for harmful effects of drugs before clinical trials, the technique used to create the model could be used to grow muscle tissue from patients with muscle diseases. This would help researchers and doctors to better understand the patient's condition and potentially develop more efficient therapies. Also, the technique could be eventually developed to grow healthy muscle tissue to implant in patients who have been injured.

to investigate biochemical changes or cell–matrix interactions that can be critical in different pathologies including muscle dystrophies and wasting disorders (*Ciciliot et al., 2013*). While three-dimensional (3D) culture models of rodent skeletal muscle have measureable contractile force (*Dennis and Kosnik, 2000*; *Huang et al., 2005*; *Hinds et al., 2011*; *Juhas et al., 2014*; *Vandenburgh et al., 2008*) and can be applied to drug testing (*Vandenburgh et al., 2008*) and disease modeling (*Lee and Vandenburgh, 2013*), in vitro 3D systems using primary human myoblasts rely on measurements of passive force (*Powell et al., 2002*; *Moon du et al., 2008*; *Mudera et al., 2010*) which is not specific to functional skeletal muscle.

Here, we describe a biomimetic human skeletal muscle culture system ('myobundle') amenable to studies of contractile function and biochemical changes in response to a wide range of stimuli. Conditions for primary myogenic cell expansion and 3D tissue formation were optimized to reproducibly obtain contractile myobundles consisting of aligned, cross-striated myofibers and a pool of cells expressing the satellite cell marker Pax7. In response to electrical and pharmacological stimuli, myobundles exhibited forceful contractions and calcium transients which could be non-invasively measured to track physiological responses and functional maturation over time. Reproducible functional characteristics were obtained using cells from nine different donors and one commercial source. Similar to clinical outcomes in humans, when pharmaceutically challenged, myobundles experienced enhanced contractile performance in response to a steroid-like substance, underwent autophagic myopathy following administration of an anti-malarial agent, and exhibited statin-induced weakness and lipid accumulation.

## Results

### Structure and composition of myobundles

Myogenic cells were isolated from human muscle biopsies and expanded for 3–5 passages, when they contained a significant fraction of muscle precursors positive for desmin and MyoD (*Figure 1—figure supplement 1*). Engineered human skeletal muscle 'myobundles' were generated using a hydrogel molding technique (*Figure 1A*, *Figure 1—figure supplement 2*) we developed for rodent cells (*Hinds et al., 2011*; *Juhas et al., 2014*). Following hydrogel compaction for 3–5 days, low serum media was applied to induce myofiber formation and differentiation. After an additional 3–5 days, the myobundles began to spontaneously twitch (*Video 1*), which was previously reported only in rodent 3D muscle constructs (*Dennis and Kosnik, 2000*). After 2-week culture, the myobundles contained densely packed and aligned myofibers embedded in a laminin-rich matrix (*Figure 1B*) and surrounded at the periphery by vimentin$^+$ fibroblasts (*Figure 1—figure supplement 3A–C*). Mature structure of the myofibers was evident by the expression of myosin heavy chain (MYH), sarcomeric alpha-actinin (SAA) cross-striations, and multiple myogenin$^+$ nuclei (*Figure 1C–E* and *Figure 1—figure supplement 2B–C*). Of functional importance, acetylcholine receptors, which are necessary for neuromuscular junction formation, were present at the myofiber surface (*Figure 1F*). While the majority of expanded myogenic cells fused to form myofibers, a fraction of cells continued to express the satellite cell marker Pax7 (*Figure 1G*), suggesting regenerative capacity as described in a rat culture model (*Juhas et al., 2014*). With time in culture, structural maturation of myobundles was evident from the progressive increase in myofiber diameter (13.5 ± 1.5 μm and 21.8 ± 2.8 μm at 1 and 4 weeks of culture, *Figure 1H*, *Figure 1—figure supplement 3D*) and expression of the muscle-specific proteins (MYH, SAA, and muscle creatine kinase (MCK), *Figure 1I*), while myofiber length and myonuclei number (524 ± 70 and 7 ± 3.6, respectively, at 3 weeks of differentiation) remained relatively steady with time of culture (*Figure 1—figure supplement 4*).

### Contractile force generation of myobundles

The amplitude of induced contractile force by electrical or chemical stimulation is a key parameter used to evaluate skeletal muscle function both in vivo and ex vivo on isolated muscle fibers (*Fuglevand et al., 1999*; *Bottinelli and Reggiani, 2000*). To optimize contractile force output of myobundles, myogenic cells were expanded in media containing either bFGF (*Ham et al., 1988*) or EGF (*Cheng et al., 2014*). Despite comparable myoblast purity and myofiber formation in 2D culture, myobundles made of EGF-expanded cells had superior contractile function (*Figure 2—figure supplement 1*). In addition to spontaneous contractions, myobundles also contracted in response to electrical stimulation (*Video 2*) and, similar to native muscle (*Rassier et al., 1999*), exhibited stronger contraction with an increase in stimulation frequency and myobundle length (*Figure 2A–B*, *Figure 2—figure supplement 2*). In concert with observed structural maturation, amplitudes of twitch and tetanus force in myobundles increased over 4 weeks in culture (*Figure 2C*), while twitch kinetics remained unchanged (*Figure 2D*).

To evaluate the robustness of the developed methodology, we expanded and utilized cells from nine donor muscle samples (obtained by needle biopsy or surgical waste) and one commercially available myoblast source (Lonza). Expanded cells from all ten sources formed functional myobundles that contracted in response to electrical stimulation with an average specific force of 2.1 ± 0.9 mN/mm$^2$ and 7.0 ± 2.2 mN/mm$^2$ for twitch and tetanus, respectively (*Figure 2E*). The average tetanus force was similar to values measured in fetal human muscle (*Racca et al., 2013*) and an order of magnitude lower than values reported for adult muscle (*Racca et al., 2013*; *Cheng et al., 2014*), while average tetanus-to-twitch ratio (3.5 ± 0.8, *Figure 2E*) was within the normal adult range (*Cheng et al., 2014*). The kinetic parameters of twitch contraction were also evaluated for each donor sample (*Figure 2F*) and were on average twofold slower than those of adult human muscle (*Fuglevand et al., 1999*) and comparable to those of single in vitro cultured human myotubes (*Smith et al., 2014*).

### Calcium handling of myobundles

To expand the utility of the myobundle platform, we incorporated a capability for non-invasive real-time monitoring of calcium transients in myobundles as calcium handling is critical to normal muscle function and can be affected by pathological conditions including dystrophic disorders and malignant hyperthermia (*Berchtold et al., 2000*). Expanded myogenic cells were lentivirally transduced with a calcium indicator, GCaMP6 (*Chen et al., 2013*), driven by a muscle-specific promoter, MHCK7 (*Salva et al., 2007*), prior to

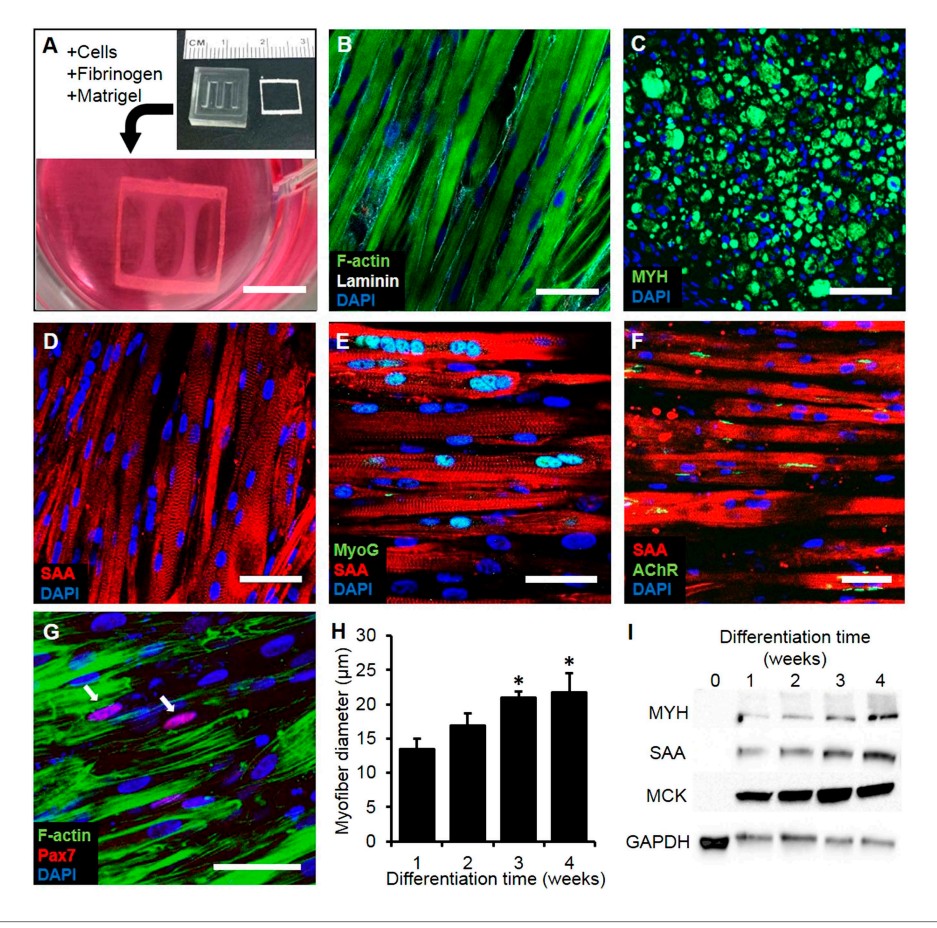

**Figure 1**. Structure and cellular composition of myobundles. (**A**) Human myogenic precursors were cast within a fibrin/matrigel matrix in PDMS molds and anchored to nylon frames. Once compacted, frames with myobundles were removed for free-floating culture. (**B**) F-actin[+] myofibers shown within 2-week myobundles are aligned and surrounded by laminin. (**C**) Transverse myobundle cross-section showing dense, uniformly distributed myosin heavy chain (MYH) expressing myofibers. (**D–F**) Aligned myofibers within myobundle show striated pattern of the contractile protein sarcomeric α-actinin (SAA) (**D**), myogenin (MyoG) positive nuclei (**E**), and bungarotoxin-labeled acetylcholine receptors (AChR) (**F**). (**G**) Pax7[+] cells (arrows) are found abutting myofibers suggesting regenerative potential. (**H**) Myofiber diameter increases with time in culture, with significant enhancement at 3 and 4 weeks vs 1 week (*p < 0.05, N = 4 donors, n > 10 myofibers per myobundle). (**I**) Structural maturation is also evident from increased expression of muscle markers MYH, SAA, and muscle creatine kinase (MCK). Scale bars: (**B–F**) scale = 50 μm, (**G**) scale = 25 μm.

The following figure supplements are available for figure 1:

**Figure supplement 1**. Myogenicity of donor cells during expansion.

**Figure supplement 2**. Schematic of myobundle fabrication.

**Figure supplement 3**. Characterization of myobundle architecture.

**Figure supplement 4**. Characterization of myofiber length and myonuclei number.

myobundle formation. As a result, robust expression of GCaMP6 in differentiated myofibers (*Figure 3A*) allowed detection of both spontaneous and electrically stimulated calcium transients in myobundles (*Figure 3B*, *Video 3*) under a variety of conditions. In response to 10 Hz (tetanic) vs single (twitch) stimuli, the amplitude of calcium transients increased (*Figure 3C–D*), as measured by normalized change in fluorescence intensity (ΔF/F), similar to the increase in contractile force with tetanic stimulation. Additionally,

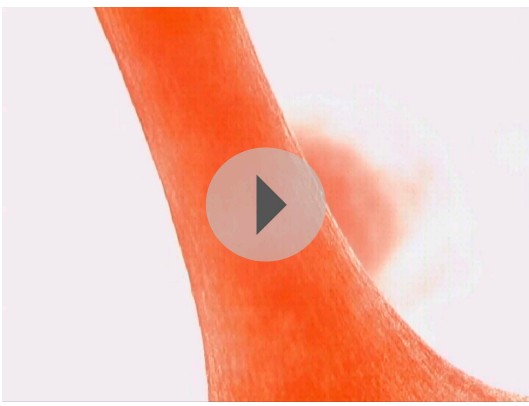

**Video 1**. Spontaneous contractions of human myobundles. Following 3–5 days of differentiation within the hydrogel construct, myofibers began spontaneously contracting. These contractions typically last for a few days, and are rarely seen beyond 2 weeks following differentiation. Video is shown in real time for 26 s duration and at field of view of 2 × 1.5 mm then 0.8 × 0.6 mm.

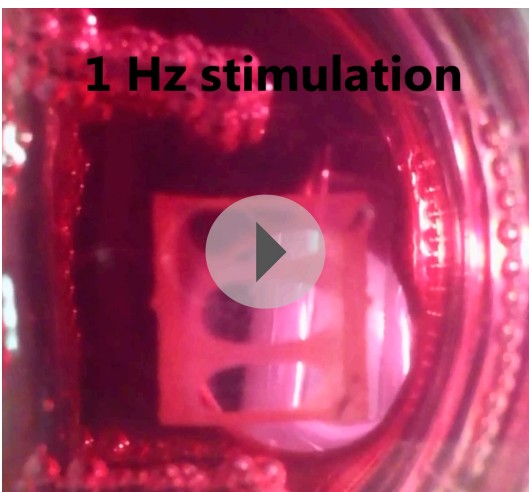

**Video 2**. Stimulated contractions of human myobundles. Myobundles respond to electrical stimulation by forceful contraction. Here, a myobundle pair is contracting in concert with 1 Hz electrical stimulation with enough force to bend the frame on which it is attached. Video is shown in real time for 17 s duration and at field of view of 25 × 25 mm.

with longer time in culture, calcium transient amplitude increased (*Figure 3D*) and correlated with the contractile force amplitude measured in the same bundles (*Figure 3—figure supplement 1*).

We further tested the functionality of calcium-handling machinery in myobundles by biochemical stimulation with caffeine and acetylcholine (ACh). By opening of ryanodine receptors, caffeine is known to generate concentration-dependent calcium release and contraction in skeletal muscle (*Moulds and Denborough, 1974*; *Berchtold et al., 2000*), as was observed in myobundles from different donors (*Figure 3—figure supplement 2*, *Video 4*). ACh is released at the neuromuscular junction to stimulate muscle contraction via opening of ligand-gated Na/K-permeable channels and voltage-gated Ca channels, while ACh receptors are a target of different muscle relaxants and toxins (*Kalamida et al., 2007*). The degree of calcium release in response to a bolus of 10 mM ACh (*Figure 3E*, *Video 5*) was comparable to that from electrically stimulated tetanus (*Figure 3F*) and unchanged throughout the entire culture period (*Figure 3—figure supplement 3A*). Tubocurarine, a muscle relaxant, blocked ACh (but not electrically) induced calcium transients (*Figure 3F*) and contractions (*Figure 3—figure supplement 3B–C*), mimicking the neuromuscular block observed in vivo (*Secher et al., 1982*).

## Drug testing of myobundles

We evaluated the potential application of myobundles as a preclinical test bed by studying their responses to three classes of pharmaceutical agents with a broad range of known effects. Statins are widely prescribed to prevent coronary artery disease, however even at normal doses some of them can induce significant myopathic weakness and rhabdomyolysis after as early as 2 weeks of use (*Dobkin, 2005*; *Thompson et al., 2006*). We tested the effects of lovastatin and cerivastatin at their clinically-relevant dose ranges (100-fold higher for lovastatin due its lower bioavailability and bioactivity [*Kantola et al., 1998*; *Shitara and Sugiyama, 2006*]) encompassing both maximum therapeutic blood serum concentrations and higher doses known to accelerate myopathic induction (*Dobkin, 2005*). In our studies, 2-week application of each statin was well tolerated in the myobundles derived from two of three donors at their respective therapeutic doses, while higher doses induced significant contractile weakness in the myobundles from all donors (*Figure 4A–B*). Unlike human myobundles that replicated clinical response, murine engineered muscle tissues in previous studies exhibited a sharp decrease in contractile function even at the lowest statin dose tested (*Vandenburgh et al., 2008*). Human myobundles also recapitulated the histopathology of statin-associated myopathy characterized by dose-dependent lipid accumulation (*Thompson et al., 2006*) (*Figure 4C*).

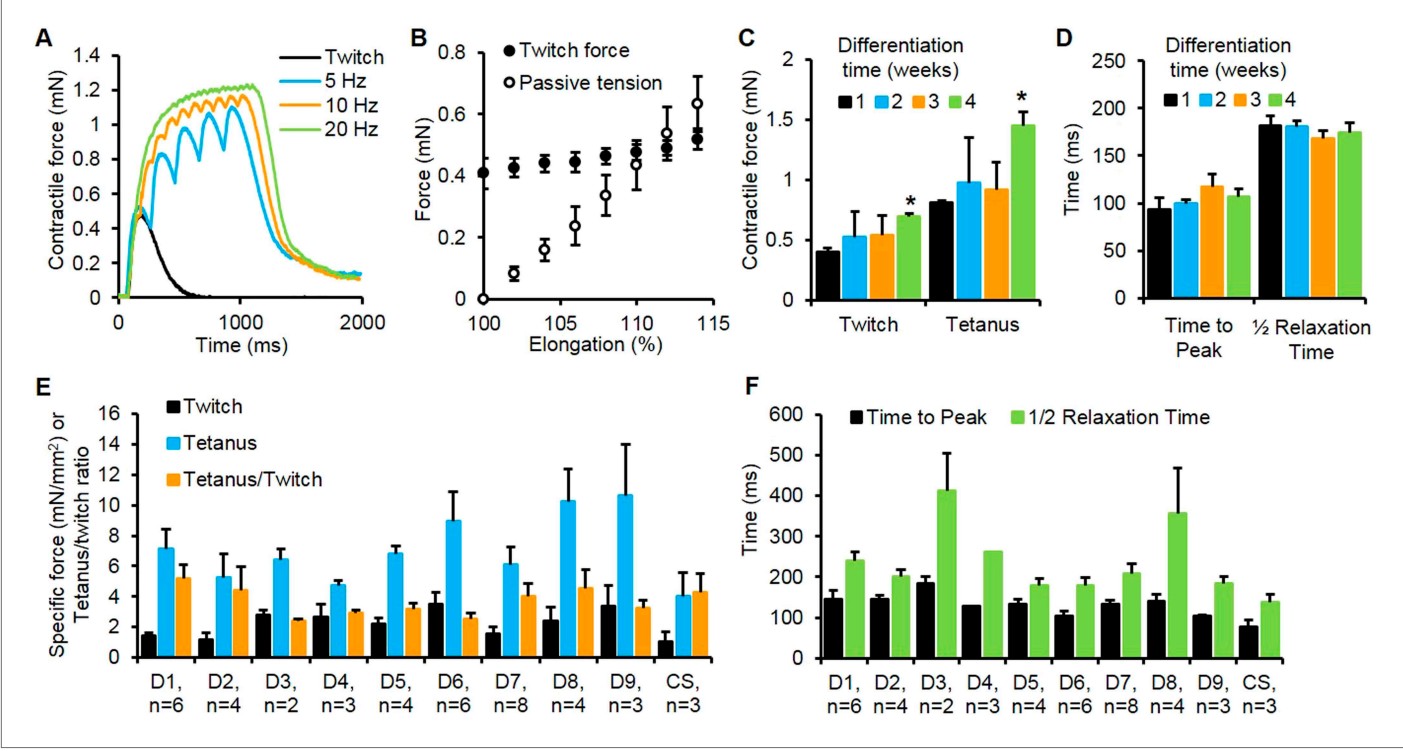

**Figure 2**. Contractile function of myobundles. (**A**) Representative contractile force traces of a 3-week myobundle showing fusion of individual twitches into a stronger tetanic contraction induced by increased stimulation frequency. (**B**) Representative increase in both contractile (active) force and passive tension with increase in myobundle length for one donor at 3 weeks in culture (n = 3 myobundles). (**C**) Twitch and tetanus forces increase over time in culture with significant enhancement at 4 weeks vs 1 week (*p < 0.05, n = 4 myobundles). (**D**) Kinetics of twitch rise and relaxation do not vary over 4 weeks in culture (n = 4 myobundles). (**E**) Specific twitch and tetanus force and tetanus-to-twitch ratio for different cell sources (D1–D9, donors 1–9; CS, commercial source, Lonza). (**F**) Kinetics of twitch response for different cell sources.

The following figure supplements are available for figure 2:

**Figure supplement 1**. Optimization of myogenic cell expansion using two different media.

**Figure supplement 2**. Force-frequency relationship of myobundles.

We also challenged myobundles with the anti-malarial agent chloroquine for 1 week to evaluate its effects on autophagy, a conserved lysosomal pathway in both physiological and pathological conditions (*Shintani and Klionsky, 2004*). With increasing doses of chloroquine, myobundles from all donors exhibited a decrease in contractile force generation (*Figure 4D*), which was associated with the autophagic buildup marked by conversion of LC3B-I to LC3B-II and a decrease in the expression of the contractile protein SAA (*Figure 4E–F*). These outcomes were consistent with autophagic-related myopathy seen in humans treated with chloroquine (*Shintani and Klionsky, 2004*; *Lee et al., 2012*). Similar biochemical responses to chloroquine including accumulation of LC3B-II and reduction of contractile proteins was also observed in 2D cultures (*Figure 4—figure supplement 1*).

Clenbuterol is a β2-adrenergic agonist with both short and long-term concentration-dependent effects on muscle, improving contractile force and hypertrophy at low concentrations, while inducing apoptosis and necrosis at high concentrations (*Burniston et al., 2006*). Clenbuterol and other β-agonists are under investigation for prevention of muscle wasting (*Ryall and Lynch, 2008*), however, species-dependent differences in their anabolic effects limit the usefulness of preclinical animal studies (*Chen and Alway, 2000*). In our studies, both acute and chronic application of clenbuterol induced an in vivo-like biphasic dose-dependent effect on contractile force generation of myobundles (*Figure 4G–H*) with the typical anabolic response and stronger contractions at 0.1 μM and diminished contractile response above 1 μM. The observed positive inotropic effect of 0.1 μM clenbuterol was partially attributed

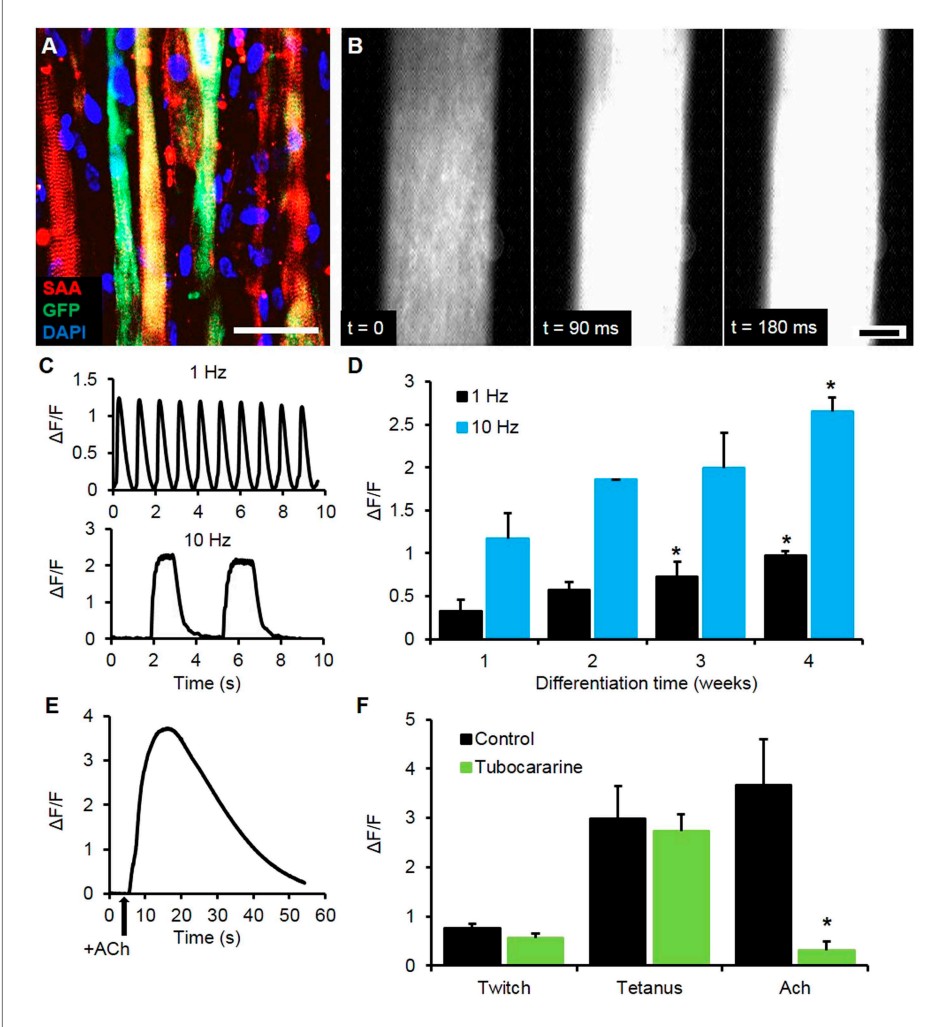

**Figure 3**. Calcium handling of myobundles. (**A**) Myofiber-specific expression of GCaMP6 in lentivirally transduced myobundles. SAA, sarcomeric α-actinin (scale bar = 50 µm). (**B**) Time course of GCaMP6 fluorescence during a single electrically stimulated twitch (scale bar = 200 µm). (**C**) Representative fluorescence traces from 1 Hz and 10 Hz stimulations of 2-week old myobundles. (**D**) Amplitude of electrically stimulated calcium transient increases with time of culture and myobundle maturation (*$p < 0.05$ vs 1 week, n = 4 myobundles). (**E**) Representative fluorescence trace of acetylcholine (ACh, 100 mM bolus) stimulated calcium release in a 2-week myobundle. (**F**) ACh receptor blocker tubocurarine (25 µM) specifically and significantly reduces ACh induced calcium release without affecting electrically stimulated calcium transients (*$p < 0.05$, n = 5 myobundles). Note that amplitude of Ach-induced calcium release is similar to that of calcium transient induced by tetanic (10 Hz) electrical stimulation.

The following figure supplements are available for figure 3:

**Figure supplement 1**. Correlation of contractile force and calcium transients.

**Figure supplement 2**. Caffeine induced calcium transients.

**Figure supplement 3**. Myobundle response to acetylcholine.

to myofiber hypertrophy (*Figure 4I*) and was confirmed in myobundles from multiple donors, resulting in an average force increase of 43.2 ± 10.8% (*Figure 4—figure supplement 2*). Collectively, these results confirm the functional similarity of myobundles to human muscle tissue and validate their potential use in the future predictive studies of muscle physiology.

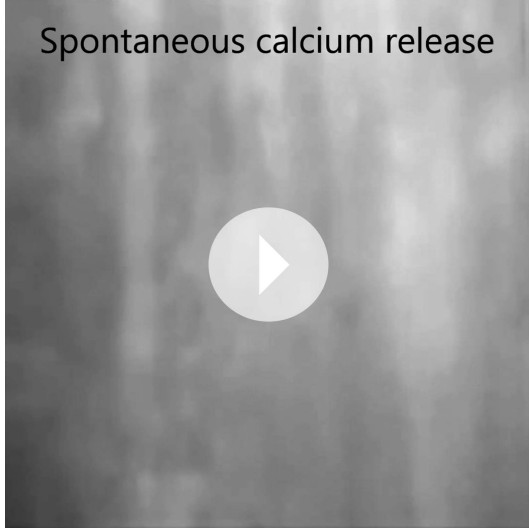

**Video 3**. GCaMP6 reported calcium release of human myobundles. Myobundles formed using myogenic precursors that were lentivirally transduced with a GCaMP6 calcium reporter contain myofibers that produce fluorescence signal in response to calcium release. Here, myobundles show calcium release in response to both twitch (1 Hz) and tetanus (10 Hz) electrical stimulation. Video is shown in real time for 36 s duration and at field of view of 500 × 500 µm.

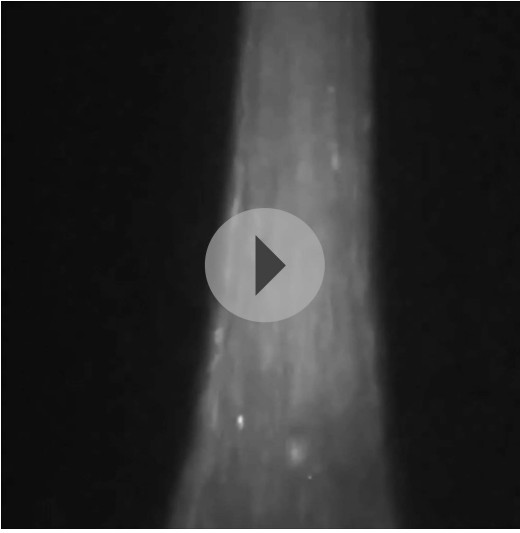

**Video 4**. Caffeine induced calcium release in human myobundles. Similar to human skeletal muscle, application of caffeine to myobundles induces calcium release via the ryanodine receptors. This lease to an increase in fluorescence from the GCaMP calcium reporter. Video is shown in real time for 66 s duration and at field of view of 2 × 2 mm.

## Discussion

We described the development and validation of the 'myobundle', a biomimetic human skeletal muscle culture platform for clinically relevant in vitro studies of muscle physiology and drug development. Myobundles recapitulate key functional aspects of human skeletal muscle including a functioning contractile apparatus, responsive acetylcholine and β2-adrenergic receptors, and physiological calcium handling, all of which are involved in pharmacological side effects in humans (*Bowes et al., 2012*). Long-term electrical and chemical responsiveness of myobundles allow for both acute and chronic physiological and pharmacological tests. Reproducibility and robustness of the system were demonstrated using biopsies from multiple donors and a commercial cell source. Correlated force generation and calcium transient responses recorded via the use of genetically encoded calcium indicators (*Chen et al., 2013*) enabled continuous optical monitoring of the relationship between stimuli and functional effects, thus bridging a significant gap in current testing methods (*Dambach and Uppal, 2012*).

Existing methods to measure contractile function of human muscle in vitro rely on acute, single-time use of intact muscle fibers isolated from patient biopsies (*Bottinelli and Reggiani, 2000*). While 2D and 3D cultures can be used to form de novo muscle fibers from human myogenic cells, existing methods fail to reproduce a comprehensive range of myofiber physiological responses, such as twitch, tetanus, and chemically induced contractions. Compared to previous 3D culture studies (*Powell et al., 2002*; *Mudera et al., 2010*), a relatively high cell density, specific hydrogel and media compositions, and dynamic culture conditions (*Juhas and Bursac, 2014*) used in our system may have all contributed to the robust formation of functional human engineered muscle. Under these conditions, the ability to generate large numbers (>1000) of contractile myobundles from a single donor biopsy allowed us to perform traditional physiological and biochemical measurements in both acute and chronic settings and for multiple testing compounds and conditions. Electrically induced calcium transients and contractions (twitch and tetanus) as well as physiological responses to increase in muscle length and stimulation frequency (*Rassier et al., 1999*; *Cheng et al., 2014*) were reproducibly recorded in myobundles from ten donors. The specific blockade of acetylcholine-induced but not electrically-induced calcium release by the muscle relaxant and acetylcholine receptor blocker tubocurarine

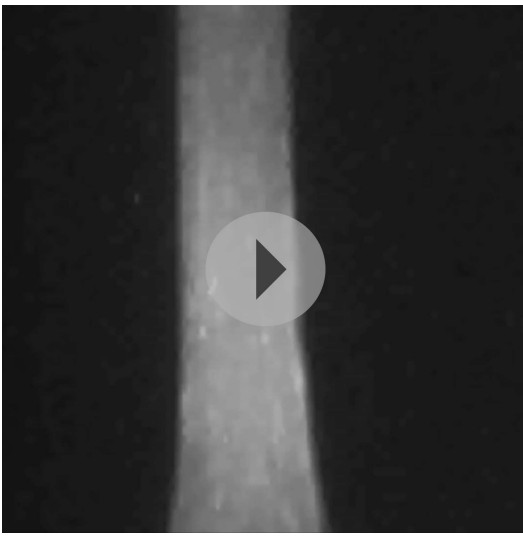

**Video 5**. Acetylcholine induced calcium release from human myobundles. Function of myobundle acetylcholine receptors was confirmed by GCaMP6 detected calcium release in response to a bolus of 10 mM acetylcholine. Myobundle response to acetylcoline was significantly blocked by the muscle relaxant tubocurarine similar to that observed clinically in human skeletal muscle. Video is shown in real time for 75 s duration and at field of view of 2 × 2 mm.

mimicked responses seen in human studies (*Secher et al., 1982*). Along with dose-dependent increase in calcium transient amplitude by caffeine, these experiments demonstrate that myofibers formed within the 3D myobundle culture environment exhibited intact excitation-contraction coupling and physiological responsiveness to both chemical and electrical stimuli. While repeated non-invasive interrogation of myobundle function was limited to calcium imaging, integration of smaller size myobundles with high-throughput force testing assays should be feasible as demonstrated for mouse cells (*Vandenburgh et al., 2008*).

The utility of myobundles as a preclinical drug testing platform was evaluated by measuring contractile and biochemical responses to statins, chloroquine, and clenbuterol. Statin myopathy is a common side effect that has been reported for all currently available statins (*Dobkin, 2005*; *Thompson et al., 2006*). Similar to clinical reports, human myobundles showed higher sensitivity to cerivastatin than lovastatin (*Shitara and Sugiyama, 2006*) and at excessive statin concentrations displayed progressive weakness and lipid accumulation, suggestive of equivalent mechanisms of action in vitro and in vivo. The use of myobundles allowed direct comparison of similar pharmaceuticals on the same patient or cohort, previously recommended for but unavailable for statins due to the variations among clinical trials and under-reporting of symptoms (*Dobkin, 2005*; *Thompson et al., 2006*). In response to an anti-malarial agent, chloroquine, myobundles showed induction of autophagic myopathy also observed in native muscle (*Shintani and Klionsky, 2004*), thus providing a potential functional screen for non-toxic modulators of autophagy. We also tested the acute and chronic responses of myobundles to β2-adrenergic agonist clenbuterol and observed myofiber hypertrophy and increased contractile strength at low clenbuterol doses followed by muscle weakness at higher doses, consistent with previous animal and human studies (*Ryall and Lynch, 2008*). Currently, binding affinity to β2-adrenergic receptors is one of the standard tests for drug specificity (*Bowes et al., 2012*) and is also a potential target for therapies in muscle wasting disorders (*Ryall and Lynch, 2008*). Overall, these results suggest that myobundles closely mimic the functional responses of native human muscle through multiple signaling pathways and could provide a pre-clinical assay for predictive screening of novel therapeutics for a broad range of muscle-related disorders.

Our in vitro model of human skeletal muscle provides a tool for improved predictive pharmacological testing and a potential alternative to costly animal studies. Non-destructive, real-time measurement of function such as calcium handling shown here could be combined with other optically-based assays (*Kleinstreuer et al., 2014*) to elucidate mechanisms of drug action. The ability to measure and quantify functional endpoints in myobundles in a population- or patient-specific manner allows construction of pharmacological time- and dose–response curves previously not available for human skeletal muscle. The myobundles may be integrated with other established human micro-organ systems such as liver or heart for more predictive body-on-chip toxicology studies (*Bhatia and Ingber, 2014*). Functional acetylcholine receptors within myobundles are integral to studies involving the neuromuscular junctions and necessary for potential implantation of such tissues to repair muscle dysfunction or loss. Eventual applications of myobundle platform using patient-derived cells to model functional deficits observed in different muscle pathologies may allow development of more efficacious therapies and safe translation to clinics.

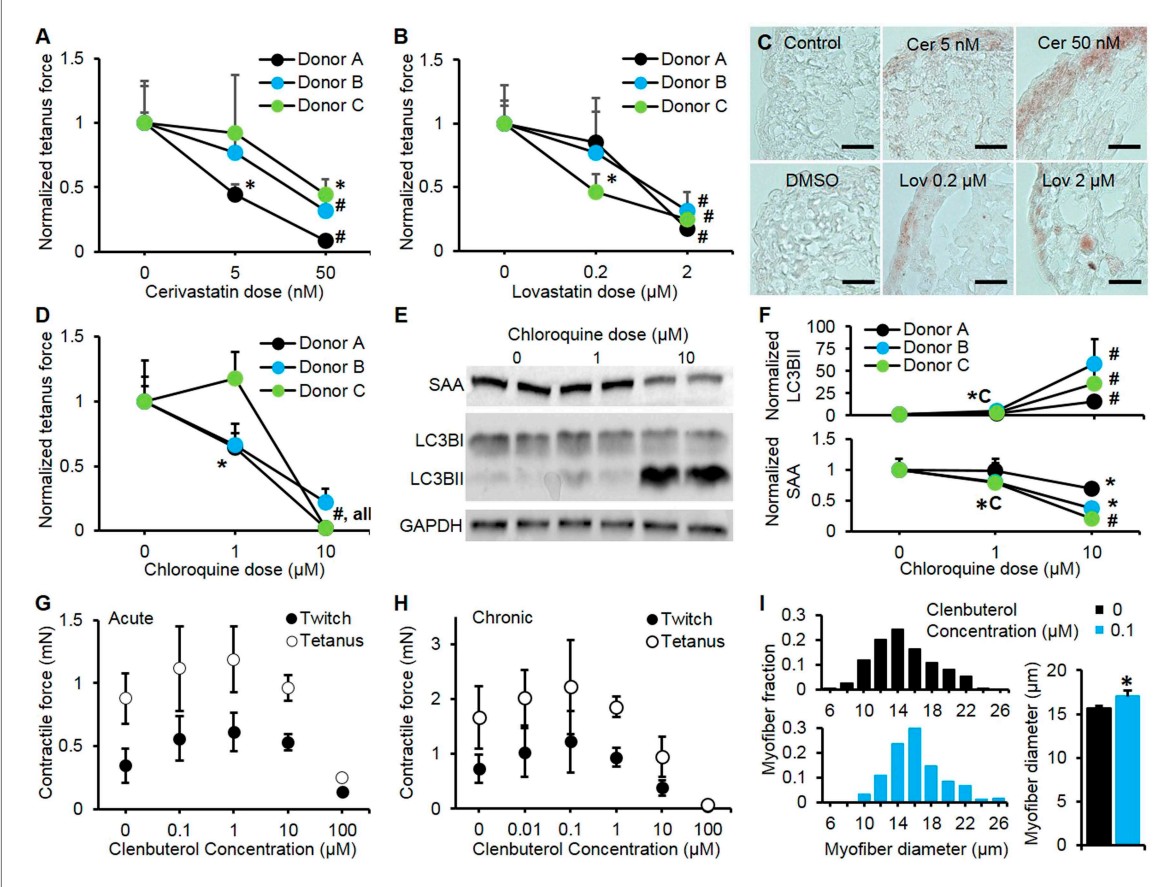

**Figure 4**. Pharmacological validation of myobundles. (**A** and **B**) 2-week application of cerivastatin (**A**) and lovastatin (**B**) at increasing doses significantly reduced tetanus force, normalized to untreated or vehicle treated (DMSO for Lovastatin) control (n = 4 myobundles per donor). (**C**) Accumulation of lipids in myobundles, evaluated by Oil Red O stain, was absent from controls, moderate at lower concentrations, and considerable at higher concentrations of both statins (scale bar = 50 μm). (**D**–**F**) 1-week exposure of myobundles to chloroquine resulted in dose-dependent decrease of contractile force (n = 4 myobundles per donor) (**D**) as well as increased expression of the autophagic pathway marker LC3B-II and decreased expression of contractile protein sarcomeric α-actinin (SAA) (**E**–**F**, n = 4 myobundles per donor). (**A**–**F**) (*p < 0.05 vs 0 μM, #p < 0.05 vs all other concentrations). (**G**) Acute (30-min) and (**H**) chronic (2-week) application of clenbuterol to myobundles (shown in different donors) results in a dose-dependent increase in contractile force with peak effects observed at 1 μM (acute) and 0.1 μM (chronic) and significant reduction in force generation observed at 100 μM (acute, n = 3 myobundles; chronic, n = 4 myobundles). (**I**) Chronic administration of 0.1 μM Clenbuterol induced hypertrophy of myofibers as evident from a rightward shift in their diameter distribution and significant increase in the average myofiber diameter (untreated, 15.7 ± 0.3 μm vs 0.1 μM clenbuterol, 17.1 ± 0.6 μm, *p < 0.05, n ≥ 55 myofibers per myobundle, pooled for 3 myobundles).

The following figure supplements are available for figure 4:

**Figure supplement 1**. Biochemical responses of human 2D myotube and 3D myobundle cultures to chloroquine.

**Figure supplement 2**. Improved myobundle function following clenbuterol treatment.

# Materials and methods

## Preparation of myogenic cells

Human skeletal muscle samples were obtained through standard needle biopsy or surgical waste under Duke University IRB approved protocols. Nine donor samples were expanded by outgrowth similar to methods previously described (*Blau and Webster, 1981*). Briefly, muscle samples were minced, washed in PBS, and enzymatically digested in 0.05% trypsin for 30 min. Muscle was collected by centrifugation, pre-plated for 2 hr, and transferred to a matrigel (BD Biosciences, San Jose, CA) coated flask for attachment. Cells were expanded in skeletal muscle growth media containing low

glucose DMEM (Gibco Life Technologies, Grand Island, NY), supplements purchased from Lonza, Basel, CH (EGF, fetuin, dexamethasone, and gentamicin without insulin), and supplemented with 10% fetal calf serum as previously described (*Cheng et al., 2014*). A second growth media containing 5 ng/ml bFGF and 20% fetal calf serum was used during optimization as it was previously shown to improve expansion of myogenic cells (*Ham et al., 1988*). Myogenic cells were either cryopreserved in 90% growth medium with 5% fetal calf serum and 5% DMSO at passage 1 or 2 then used at passage 3–5 for the generation of myobundles or staining. A sample of primary human skeletal myoblasts from additional donor was purchased from Lonza for comparison.

For calcium imaging studies, expanded myogenic cells were transduced with a lentiviral vector encoding the fluorescent calcium reporter GCaMP6 (*Chen et al., 2013*) driven by a myosin heavy chain-creatine kinase promoter MHCK7 (*Salva et al., 2007*) for muscle specific expression.

For the measurements of myofiber length and nuclei number, 5% of myogenic cells used for myobundle formation were transduced with a lentiviral vector encoding MHCK7 driven GFP (*Li et al., 2011*). This allowed the visualization and measurement of individual GFP+ myotubes within myobundles using immunostaining and confocal microscopy.

## Fabrication of human myobundles

Myobundles were formed by modifying our previously published methods for engineered rodent muscle tissues (*Hinds et al., 2011*; *Juhas et al., 2014*) (*Figure 1—figure supplement 2*). Expanded myogenic cells were dissociated in 0.025% trypsin-EDTA to a single cell suspension and encapsulated in a fibrinogen (Akron, Boca Raton, FL) and matrigel solution on laser cut Cerex frames (9.2 × 9.5 mm outer dimensions, 6.8 × 8.3 mm inner dimensions) within PDMS molds (cast from Teflon masters and pre-treated with pluronic) at $15 \times 10^6$ cells/ml ($7.5 \times 10^5$ cells per myobundle). Specifically, a cell solution ($7.5 \times 10^5$ cells in 17.2 µl media per bundle + 2 µl of 50 unit/ml thrombin in 0.1% BSA in PBS [Sigma, St. Louis, MO]) and a gelling solution (11 µl media + 10 µl Matrigel + 10 µl of 20 mg/ml Fibrinogen in DMEM) were prepared in separate vials on ice for up to six myobundles per vial. Gelling solution was added to the cell solution and mixed thoroughly then each bundle was individual pipetted within the PDMS mold and onto the frame. The cell/hydrogel mixture was polymerized for 30 min at 37°C followed by incubation in growth media containing 1.5 mg/ml 6-aminocaproic acid (ACA, Sigma). Myobundles were kept in growth media during gel compaction (3–5 days) and then switched to low glucose DMEM with 2% horse serum (Hyclone, Logan, UT), 2 mg/ml ACA and 10 µg/ml insulin (Sigma). Frames were removed from molds at the time of switch to low serum medium and cultured dynamically in suspension for an additional 1–4 weeks. Starting from a 50 mg donor biopsy, typical cell expansion for 5 passages can allow generation of at least 1000 myobundles with a total mass of >5 g, representing a >100-fold amplification of muscle mass when going from native to engineered tissue system.

All drugs were purchased from Sigma. Clenbuterol hydrochloride, chloroquine phosphate, and ceriv-astatin sodium salt hydrate were prepared at 1000× stock solutions in PBS (control) and sterile-filtered for use. Lovastatin was prepared as a 10,000× stock solution in DMSO in which case DMSO was used as vehicle control. Drugs studies in myobundles or 2D cultures were initiated after 1 week of differentiation. Myobundles were replenished with fresh media and drug each day to maintain drug concentration.

## Measurement of contractile force

Electrically or chemically stimulated contractile force generation in myobundles was measured using a custom force measurement set-up as previously described (*Hinds et al., 2011*; *Juhas et al., 2014*). Briefly, single myobundles on a frame were transferred to the bath of the force measurement set-up, maintained at 37°C. One end of the bundle was secured by a pin to an immobile PDMS block and the other end was attached to a PDMS float connected to the force transducer mounted on a computer-controlled motorized linear actuator (Thor Labs, Newton, NJ). The sides of the frame were cut to allow myobundle stretch by the actuator and isometric measurement of contractile force. Initially, the myobun-dle was set to its baseline length using the motorized linear actuator. To assess the force-length relation-ship, myobundle was stretched by 2% of its culture length then stimulated by a 40 V/cm, 10 ms electrical pulse using a pair of platinum electrodes and the twitch force was recorded. At 12% stretch, 1 s long stimulations at 5, 10, and 20 Hz were applied and the subsequent contractile force was recorded to assess the force-frequency relationship. Contractile force traces were analyzed for peak twitch or tetanus force, time to peak twitch, and half relaxation time using a custom MATLAB program (*Source code 1*). For studies with acetylcholine, 60 µl of drug solution was added to the 6 ml bath at t = 5 s of recording.

## Imaging of calcium transients

Myobundles expressing the MHCK7-GcaMP6 reporter were non-destructively monitored for calcium transients following differentiation. A live imaging chamber with heated enclosure was used to maintain cells in physiological conditions during recording. Bundles were placed in sterile tyrode's solution in a custom-designed glass-bottom bath containing electrodes for stimulation. Video-images were acquired using an Andor iXon camera affixed to a Nikon microscope with a FITC filter and either 4× or 10× objective. During studies with caffeine and acetylcholine, 60 µl of drug solution was added to the bath at t = 5 s of recording. Video was analyzed using Andor Solis software and relative changes in fluorescence signal were calculated by $\Delta F/F = (Peak-Trough)/(Trough-Background)$ as previously described (*Juhas et al., 2014*).

## Immunohistochemistry

Cells were fixed in 4% paraformaldehyde in PBS for 10 min and myobundles were fixed in 2% paraformaldehyde in PBS overnight at 4°C. Following fixation, samples were washed in PBS then blocked in 5% chick serum with 0.2% Triton-X 100. The following primary antibodies were used for tissue characterization: desmin (SCBT, Dallas, TX, 1:200), anti-GFP (Life Technologies, 1:200), laminin (Abcam, Cambridge, MA, 1:200), muscle creatine kinase (SCBT, 1:100), MyoD (BD, 1:100), myogenin (SCBT, 1:100), myosin heavy chain 1/2/4/6 (SCBT, 1:100), Pax7 (DSHB, Iowa City, IA, 1:50), sarcomeric α-actinin (Sigma, 1:200), and vimentin (Sigma, 1:200). Corresponding fluorescently labeled secondary antibodies (1:200), α-bungarotoxin (1:100), and phalloidin (1:200) were purchased from Life Technologies. Oil Red O staining was performed using standard protocols on cryosections of myobundles fixed in 4% paraformaldehyde. Hematoxylin and eosin stain was performed on paraffin embedded sections of 2% paraformaldehyde fixed myobundles using Harris modified hematoxylin (Sigma) and Eosin Y (Sigma). Images were acquired using a Zeiss 510 inverted confocal microscope and analyzed using LSM Image Software. Mosaic images for fiber length measurements were generated using Mosaic J in FIJI.

## Western blotting

Cell or myobundle protein was isolated in RIPA lysis and extraction buffer with protease inhibitor (Sigma). Protein concentration was determined using BCA assay (Pierce of Thermo Scientific, Rockford, IL) according to manufacturer's instructions. Western blot was performed using Bio-Rad Mini-PROTEAN gels and the Mini-PROTEAN Tetra cell, Mini Trans-blot module (Bio-Rad, Hercules, CA). The following primary antibodies were used for detection: GAPDH (SCBT, 1:500), LC3 (Cell Signaling, 1:200), muscle creatine kinase (SCBT, 1:200), myosin heavy chain 1/2/4/6 (SCBT, 1:200), and sarcomeric alpha-actinin (Sigma, 1:200). HRP conjugated anti-mouse (1:20,000) and anti-goat (1:5000) antibody were purchased from Sigma, and HRP conjugated anti-rabbit was purchased from SCBT (1:5000). Chemiluminescence was performed using Clarity Western ECL substrate (Bio-Rad). Images were acquired using a Bio-Rad Chemidoc and analyzed using ImageJ.

## Statistics

Results are presents as mean ± SD. Statistical significance was determined by unpaired t-test or one-way ANOVA with post-hoc Bonferroni–Holm test. $p < 0.05$ was considered statistically significant.

## Acknowledgements

We are grateful to Dr Edward Smith and Dr Robert Lark for providing muscle biopsies. We thank Dr Thomas Rando for reading of the manuscript.

## Additional information

### Funding

| Funder | Grant reference number | Author |
| --- | --- | --- |
| National Institute of Arthritis and Musculoskeletal and Skin Diseases | R01AR055226 and R01AR065873 | Nenad Bursac |
| National Institutes of Health | UH2TR000505 | George A Truskey |

The funders had no role in study design, data collection and interpretation, or the decision to submit the work for publication.

## Author contributions

LM, MJ, Conception and design, Acquisition of data, Analysis and interpretation of data, Drafting or revising the article; WEK, Conception and design, Analysis and interpretation of data, Drafting or revising the article, Contributed unpublished essential data or reagents; GAT, NB, Conception and design, Analysis and interpretation of data, Drafting or revising the article

## Ethics

Human subjects: Human skeletal muscle samples were obtained through standard needle biopsy or surgical waste under Duke University IRB approved protocols (Pro00048509 and Pro00012628).

## Additional files

### Supplementary file

• Source code 1. Source code for analysis of contractile force in MATLAB.

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
