## [Decision Letter]

Thank you for sending your work entitled “Bioengineered human myobundles mimic clinical responses of skeletal muscle to drugs” for consideration at *eLife*. Your article has been favorably evaluated by Sean Morrison (Senior editor), Amy Wagers (Reviewing editor), and 3 reviewers.

The Reviewing editor and the reviewers discussed their comments before we reached this decision, and the Reviewing editor has assembled the following comments to help you prepare a revised submission.

All of the reviewers found your work to be novel, interesting and important. They were particularly impressed with your success in developing a robust, physiologically relevant 3D model of human skeletal muscle that shows electrical responsiveness, and with the rigor of your analyses, using multiple patient biopsies instead of cell lines, and validating with three different drugs. Overall, it was felt that your results are likely to be highly influential, particularly in the growing areas of tissue engineering and patient-specific drug screening, and provide a significant advance for the field.

That said, there were a few issues that surfaced during review and discussion, which we would like you to address in a revised version. These essential revision requirements are outlined in the numbered list below. Following this list, we have included some additional, minor comments that you may wish to address as well. We look forward to receiving your revised manuscript.

1) A key determinant of the long-term impact of this work will be the degree to which other groups are able to adopt your system for drug screening and analysis. To this end, it will be essential to ensure that the methodologies are clearly and completely explained. Currently, there are important details missing (e.g., regarding the fabrication procedure, how force is measured, how and when growth factors were added, hydrogel formulation, etc.). Please provide additional necessary detail, and consider inclusion of schematic diagrams, to enable others to replicate this system.

2) Related to point 1, while you demonstrate beautifully that the myobundle system phenocopies clinical responses to drugs, to convince researchers to convert from the 'user-friendly' 2D model to your system it will be necessary to include data comparing these systems side-by-side. Certainly, as you already point out, twitch and tetanus measurements are only possible in 3D, but it is important also to include data that compares some of the other metrics you assayed to indicate drug response (e.g., myofiber diameter, autophagy, lipid accumulation, etc.) in 2D vs. 3D, for at least one of the drug treatments analyzed.

3) Your report of spontaneous contraction of human myofibers and pacing in response to electrical stimulation is a first in the field, as human myofibers notoriously fail to respond to electrical stimulation. The findings would be more impactful, however, if you could offer explanation as to why your system supports human myofiber contraction while another quite similar 3D human skeletal muscle system published by the Vandenburgh lab did not demonstrate similar success. Do you also see spontaneous twitching in 2D cultures? If yes, perhaps it has something to do with isolation and culture media formulations? If not, then what is different from the prior unsuccessful reports?

4) From Figure 1—figure supplement 2, it appears that fibers are well aligned in the core of the tissue, but splaying outwards in the outer edges. Fibers also seem quite short and do not contain many myonuclei/fibers. Together these observations cast a bit of doubt on the functional force measurements. Quantification of these parameters is needed to support the conclusions.

5) Some reviewers were concerned that the paper may overstate the relevance of this model to testing drugs in disease relevant models of skeletal muscle disease, as only healthy muscle was evaluated. Please revise the text to avoid the implication that the system is validated for studies of diseased muscle. (Please note that we are *not* asking for new experiments with diseased muscle to be included, just for a clarification of the text and conclusions.)

---

## [Author Response]

*1) A key determinant of the long-term impact of this work will be the degree to which other groups are able to adopt your system for drug screening and analysis. To this end, it will be essential to ensure that the methodologies are clearly and completely explained. Currently, there are important details missing (e.g., regarding the fabrication procedure, how force is measured, how and when growth factors were added, hydrogel formulation, etc.). Please provide additional necessary detail, and consider inclusion of schematic diagrams, to enable others to replicate this system*.

We agree with the reviewers and have significantly expanded our Methods section. Specifically, we described in more detail the cell expansion and exact contents of the growth media:

*“*Human skeletal muscle samples were obtained through standard needle biopsy or surgical waste under Duke University IRB approved protocols. Nine donor samples were expanded by outgrowth similar to methods previously described (Blau and Webster, 1981). Briefly, muscle samples were minced, washed in PBS, and enzymatically digested in 0.05% trypsin for 30 minutes. Muscle was collected by centrifugation, pre-plated for 2 hours, and transferred to a matrigel (BD) coated flask for attachment. Cells were expanded in skeletal muscle growth media containing low glucose DMEM, supplements purchased from Lonza (EGF, fetuin, dexamethasone, and gentamicin without insulin), and supplemented with 10% fetal calf serum as previously described (Cheng et al., 2014). A second growth media containing 5 ng/mL bFGF and 20% fetal calf serum was used during optimization as it was previously shown to improve expansion of myogenic cells (Ham et al., 1988). Myogenic cells were either cryopreserved in 90% growth medium with 5% fetal calf serum and 5% DMSO at passage 1 or 2 then used at passage 3-5 for the generation of myobundles or staining. A sample of primary human skeletal myoblasts from additional donor was purchased from Lonza for comparison.”

We added a new supplementary figure (Figure 1—figure supplement 2) with the schematic of myobundle fabrication procedure and with images showing the appearance of myobundles at different stages of culture. Furthermore, we described fabrication of myobundles in more detail, as follows, in the Materials and methods section:

“Myobundles were formed by modifying our previously published methods for engineered rodent muscle tissues (Hinds et al., 2011; Juhas et al., 2014) (Figure 1—figure supplement 2). Expanded myogenic cells were dissociated in 0.025% trypsin-EDTA to a single cell suspension and encapsulated in a fibrinogen (Akron) and matrigel (BD Biosciences) solution on laser cut Cerex® frames (9.2 x 9.5 mm outer dimensions, 6.8 x 8.3 mm inner dimensions) within PDMS molds (cast from Teflon masters and pretreated with pluronic) at 15x10^6^ cells/mL (7.5x10^5^ cells per myobundle). Specifically, a cell solution (7.5x10^5^ cells in 17.2 µL media per bundle + 2 µL of 50 unit/mL thrombin in 0.1% BSA in PBS (Sigma)) and a gelling solution (11 µL media + 10 µL Matrigel + 10 µL of 20 mg/mL Fibrinogen in DMEM) were prepared in separate vials on ice for up to six myobundles per vial. Gelling solution was added to the cell solution and mixed thoroughly then each bundle was individual pipetted within the PDMS mold and onto the frame. The cell/hydrogel mixture was polymerized for 30 min at 37°C followed by incubation in growth media containing 1.5 mg/mL 6-aminocaproic acid (ACA, Sigma). Myobundles were kept in growth media during gel compaction (3-5 days) and then switched to low glucose DMEM (Gibco) with 2% horse serum (Hyclone), 2 mg/mL ACA and 10 µg/mL insulin (Sigma). Frames were removed from molds at the time of switch to low serum medium and cultured dynamically in suspension for an additional 1–4 weeks.”

Additional text on the details of force measurement was added as well to the same section:

“Electrically or chemically stimulated contractile force generation in myobundles was measured using a custom force measurement set-up as previously described (Hinds et al., 2011; Juhas et al., 2014). Briefly, single myobundles on a frame were transferred to the bath of the force measurement set-up, maintained at 37°C. One end of the bundle was secured by a pin to an immobile PDMS block and the other end was attached to a PDMS float connected to the force transducer mounted on a computer-controlled motorized linear actuator (Thor Labs). The sides of the frame were cut to allow myobundle stretch by the actuator and isometric measurement of contractile force. Initially, the myobundle was set to its baseline length using the motorized linear actuator. To assess the force-length relationship, myobundle was stretched by 2% of its culture length then stimulated by a 40V/cm, 10 ms electrical pulse using a pair of platinum electrodes and the twitch force was recorded. At 12% stretch, 1 second long stimulations at 5, 10, and 20 Hz were applied and the subsequent contractile force was recorded to assess the force-frequency relationship. Contractile force traces were analyzed for peak twitch or tetanus force, time to peak twitch, and half relaxation time using a custom MATLAB program. For studies with acetylcholine, 60 µL of drug solution was added to the 6 mL bath at t = 5 sec of recording.”

We also added a line, in the same section of the text, to indicate the timing of drug additions:

“Drugs studies in myobundles or 2D cultures were initiated after one week of differentiation. Myobundles were replenished with fresh media and drug each day to maintain drug concentration.”

*2) Related to point 1, while you demonstrate beautifully that the myobundle system phenocopies clinical responses to drugs, to convince researchers to convert from the 'user-friendly' 2D model to your system it will be necessary to include data comparing these systems side-by-side. Certainly, as you already point out, twitch and tetanus measurements are only possible in 3D, but it is important also to include data that compares some of the other metrics you assayed to indicate drug response (e.g., myofiber diameter, autophagy, lipid accumulation, etc.) in 2D vs. 3D, for at least one of the drug treatments analyzed*.

To address this comment, we performed additional experiments to directly compare drug responses in 2D and 3D cultures. Specifically, cells from the same donor were seeded on 2D Matrigel coated dishes or formed into 3D myobundles. The preparations were differentiated for one week and then treated for one week with different doses of chloroquine, as in our original 3D experiments. Western blot analysis showed a similar, dose-dependent increase in expression of LC3B-II and decrease in expression of contractile proteins sarcomeric α-actinin (SAA) and myosin heavy chain (MYH) in 2D and 3D cultures. Overall, while we expectedly found similar biochemical effects of chloroquine on autophagic buildup in myofibers cultured either in a 2D or 3D environment, the 3D myobundles allowed additional functional studies compared to 2D that can correlate the biochemical findings to a functional output. These results are now shown in new Figure 4—figure supplement 1 and described in the revised Results section as follows:

“Similar biochemical responses to chloroquine including accumulation of LC3B-II and reduction of contractile proteins was also observed in 2D cultures (Figure 4—figure supplement 1).”

*3) Your report of spontaneous contraction of human myofibers and pacing in response to electrical stimulation is a first in the field, as human myofibers notoriously fail to respond to electrical stimulation. The findings would be more impactful, however, if you could offer explanation as to why your system supports human myofiber contraction while another quite similar 3D human skeletal muscle system published by the Vandenburgh lab did not demonstrate similar success. Do you also see spontaneous twitching in 2D cultures? If yes*, *perhaps it has something to do with isolation and culture media formulations? If not, then what is different from the prior unsuccessful reports?*

We did not observe spontaneous twitching of myofibers in our 2D human cultures, whereas others have shown that sporadic spontaneous twitches in 2D cultures can be observed in more complex differentiation media (Guo et al., 2014). Compared to our myobundle fabrication protocol, other 3D human muscle systems utilized collagen I-based rather than fibrin-based matrix and employed lower cell densities (5x10^6^ cells/mL (Powell et al., 2002; Powell et al., 1999), 4-8 x10^6^ cells/mL (Mudera et al., 2010)). Furthermore, a recent study with fibrin-based human muscle model has utilized a much higher (>10x) fibrinogen concentration (200 mg/mL stock solution, 57 mg/mL final concentration (Martin et al., 2013) compared to our study.

Previously, we showed that fibrinogen and matrigel concentrations are important determinants of contractile function of our 3D engineered rat muscles (Hinds et al., 2011) and that the use of fibrin gel outperformed use of collagen I gel with respect to engineered muscle function and long-term stability (Hinds et al., 2011; Bian and Bursac, 2009). In optimizing human 3D muscle system, we relied on our significant experience with 3D rat system. We tested multiple fibrinogen concentrations and starting cell densities in human system and found that 4 mg/ml fibrinogen concentration and 15x10^6^ cells/mL cell density allowed both development of robust contractile activity as well as long term stability of the engineered muscle. Myobundles made using low fibrinogen concentrations thinned quickly and were unstable. Higher fibrinogen concentrations decreased myobundle compaction rates and cell fusion leading to a reduced contractile force generation.

With regards to culture media, previous reports of 3D human muscle utilized high glucose medium for both differentiation and maintenance of cells. We used low glucose media and supplemented the differentiation media with 10 μg/mL insulin. While insulin supplementation was not completely necessary, it significantly enhanced force production (>2 fold) compared to differentiation without insulin. While adding various growth factors (e.g. IGF-1) could further improve function of myobundles, we chose to limit the cost and complexity of our differentiation media to what could be readily affordable and reproducible by others. Finally, for our myobundles, we utilized dynamic instead of static culture conditions. Similar to our recently published study in rat 3D muscle system (Juhas and Bursac; 2014), dynamic conditions additionally enhanced force producing capacity of our human engineered muscle.

Taken together, while we cannot know every detail of cell expansion and culture conditions explored by others, we believe that particular combination of high cell density, fibrinogen concentration, media formulation, and dynamic culture conditions used in our study resulted in robust contractile response of our 3D human myobundles.

These, to our belief, important differences from previous 3D studies have been now emphasized in the Discussion as follows:

“Compared to previous 3D culture studies (Powell et al., 2002; Mudera et al., 2010), a relatively high cell density, specific hydrogel and media compositions, and dynamic culture conditions (Juhas and Bursac, 2014) used in our system may have all contributed to the robust formation of functional human engineered muscle.”

*4) From*
Figure 1—figure supplement 2*, it appears that fibers are well aligned in the core of the tissue, but splaying outwards in the outer edges. Fibers also seem quite short and do not contain many myonuclei/fibers. Together these observations cast a bit of doubt on the functional force measurements. Quantification of these parameters is needed to support the conclusions*.

We acknowledge that original Figure 1—figure supplement 2 could leave such an impression. We have thus reexamined similar immunostainings from different myobundles and established that apparently splayed fibers in that panel have been an artifact of imaging. We have replaced this panel with a more representative composite confocal image showing the uniform fiber alignment throughout the myobundle. Uniform fiber alignment is also evident from Figure 1 and Figure 1—figure supplement 3. In addition, we have performed new and non-trivial experiments to quantify fiber length and number of myonuclei per fiber. Specifically, to be able to evaluate length of individual fibers within the intact myobundles, we have applied a low concentration of a lentivirus driving the GFP expression under an MHCK7 promoter to transduce 5% of myogenic cells before encapsulating them in hydrogel. This low level of GFP labeling resulted in appearance of clearly distinguishable individual GFP+ fibers within myobundles. These fibers were imaged by confocal microscopy and their length and number of myonuclei were evaluated at 1, 2, and 3 weeks post-differentiation. New Figure 1—figure supplement 4 shows the representative images of the labeled fibers and corresponding average lengths and myonuclei numbers. The measured fiber length in human myobundles is consistent with the only previous report of the fiber length in engineered muscle tissues (made of primary mouse myoblasts) that we could find (Li et al., 2011).

These data shown in new Figure 1—figure supplement 4 are now described in the revised Results section as follows:

“…while myofiber length and myonuclei number (524±70 and 7±3.6, respectively, at 3 weeks of differentiation) remained relatively steady with time of culture (Figure 1—figure supplement 4).”

The methods to assess myofiber length and myonuclei number are described in the revised Methods section as follows:

“For the measurements of myofiber length and nuclei number, 5% of myogenic cells used for myobundle formation were transduced with a lentiviral vector encoding MHCK7 driven GFP (Li et al., 2011). This allowed the visualization and measurement of individual GFP^+^ myotubes within myobundles using immunostaining and confocal microscopy.”

*5) Some reviewers were concerned that the paper may overstate the relevance of this model to testing drugs in disease relevant models of skeletal muscle disease, as only healthy muscle was evaluated. Please revise the text to avoid the implication that the system is validated for studies of diseased muscle. (Please note that we are* not *asking for new experiments with diseased muscle to be included, just for a clarification of the text and conclusions*.*)*

We have edited the text to avoid implications that our system was validated for studies of diseased muscle. Specifically, we eliminated the mentioning of disease from the following sentences:

In the Discussion: *“*In response to an anti-malarial agent, chloroquine, myobundles showed induction of autophagic myopathy also observed in native muscle (Shintani and Klionsky, 2004), thus providing a potential functional screen for non-toxic modulators of autophaghy.”

In the Results: “Collectively, these results confirm the functional similarity of myobundles to human muscle tissue and validate their potential use in the future predictive studies of muscle physiology.”

In the Discussion: “We described the development and validation of the ‘myobundle’, a biomimetic human skeletal muscle culture platform for clinically relevant in vitro studies of muscle physiology and drug development.”